# D2D Social Selection Relay Algorithm Combined with Auction Principle

**DOI:** 10.3390/s22239265

**Published:** 2022-11-28

**Authors:** Hairui Wang, Yijun Wang, Luping Tang, Yongqiang Xia

**Affiliations:** School of Electronic and Information Engineering, Changchun University of Science and Technology, Changchun 130022, China

**Keywords:** D2D, social perception, social threshold, auction algorithm

## Abstract

In a D2D (device-to-device) communication system, this paper proposes a relay selection strategy based on social perception. Firstly, the social threshold is introduced into the D2D relay network to screen and filter the potential relay users, thus effectively reducing the detection cost. Then, an auction algorithm is used to motivate the relay users to increase their transmission power. The simulation results show that the algorithm not only improves the throughput but also reduces the probability of a system outage.

## 1. Introduction

With the rapid development of mobile communication and the increases in users, it is increasingly difficult for the existing mobile network infrastructure to meet the communication requirements. To solve this problem, device-to-device (D2D) communication technology came into being. It can reduce the burden of the base station and meet the communication needs of users. When the channel fading is serious or the communication distance is long, the selection of an appropriate relay technology can improve the user experience and reduce the burden of the base station [1,2]. At present, most relay schemes focus on how to select relay nodes in the physical-layer technology. The selfish behavior of the relay user equipment is measured based on three attributes, the joint interest degree, forward history ratio, and relay physical state, and the attribute difference between the source node and relay node is calculated by using the idea of a triangular fuzzy function. Through an attribute difference comparison, the nodes with selfish behavior are excluded [3]. The asymptotic analysis of a mutual outage probability under a high signal-to-noise ratio and the estimation of its diversity order provides a secure communication environment for cellular networks and increases the throughput of the system [4]. Aiming at the problems of relay selection and channel power allocation in D2D communication, an iterative Hungarian algorithm is proposed to make the performance of the communication link better [5]. The reference [6] proposes an energy cooperation method of the D2D relay technology. When the energy of the device is insufficient, the data transmission can be completed by other devices. The reference [7] performs a relay selection by using the cross-layer relay selection scheme based on the Q-learning algorithm, which can significantly improve the transmission efficiency of the system, even under the premise of an unknown CSI. The reference [8] proposes an optimal power allocation scheme by analyzing the interference of the D2D relay communication in the full-duplex mode, so that the relay communication can provide a gain for the system transmission capacity in the full-duplex mode. In the full-duplex mode, the problem is transformed into a one-to-one weighted bipartite matching problem and the optimal solution is obtained by using the Hungarian algorithm [9]. A collection-before-transmission model is proposed for D2D communication based on an EH, and the throughput is improved [10]. The reference [11] proposes a high energy efficiency power distribution for multiple relay auxiliary device-to-device communication using an iterative and high energy efficiency relay selection algorithm. The mode selection algorithm is used to select the appropriate communication mode for cellular users [12]. The reference [13] proposes a method based on unsupervised learning, which is developed by the k-means to help select the relay nodes. For data transmission between the source node and the relay node, a random path selection protocol is proposed [14]. The source node uses reinforcement learning technology to select the appropriate one according to the environment. The references [15,16] propose a power optimization algorithm, which converts the non-convex energy efficiency optimization into a convex optimization problem for solving. On the premise of determining the number of relays, a mode selection algorithm with optimal energy efficiency is proposed, which improves the energy efficiency [17]. To solve the problem that relay nodes consume their own energy when providing forwarding data, the reference [18] proposes a heterogeneous cellular network that provides energy for D2D communication, in which the relay device collects energy from the access point and uses the collected energy for D2D communication.

The above solutions are all considered from the perspective of the physical layer and assume that all relay equipment users are willing to provide relay services. However, this ideal state is difficult to achieve in the real world, and not all device users are willing to provide relay services. To address this problem, the reference [19] proposes a relay selection method by using social networks. The physical domain and social domain of the relay are considered comprehensively, and the transmission stability of the link is calculated according to the user’s encounter history and the cooperation willingness is calculated according to the user’s intimacy, which are used as the criteria to select the relay. This method can improve the probability of a successful relay selection and reduce the burden of the cellular network. The reference [20] proposes an innovative social-aware energy-saving relay selection mechanism, which considers the hidden social relations among mobile users and can ensure that more users are willing to participate in cooperative communication. The reference [21] puts forward a kind of application in the IoT scenarios where a D2D communication optimal routing algorithm with more jump, combined with the social domain information in the algorithm, using the trust model based on sorting the trust of the D2D connection probability was derived, whereby the trust probability obtained the trusted connection between the nodes’ probability and, in the algorithm, considering the influence of the trusted connection probability. In the relay selection process, not only the social relationship between the relay node and sender but also the social relationship between the relay node and receiver are considered [22]. An optimal stopping algorithm is proposed, which not only considers the strength of the social relationship but also reduces the detection times and saves the cost by using the optimal stopping algorithm [23]. By fusing the relay selection algorithm based on distance and the relay selection algorithm based on the social relationship, a hybrid relay selection algorithm is proposed, which has a better effect [24].

In this article, we propose a new algorithm for the relay selection by using social relations. The main contributions of this work are as follows:The willingness of relay users to forward is fully considered, so we introduce the social weight to represent the willingness to cooperate. The cost of detecting the channel state of each relay link is reduced by excluding uncooperative nodes.The relay nodes that are not willing to cooperate are excluded, the overall outage probability of the system is found out from the existing nodes, and the relay nodes with a small outage probability are screened out.The auction algorithm is introduced, and the monetary incentive is used to stimulate the forwarding power of the relay users to reduce the outage probability and improve the system throughput.The proposed algorithm is simulated and compared with the existing algorithms. The results show that the proposed algorithm can reduce the outage probability of the D2D communication link and improve the throughput.

The following contents of this paper are as follows: In Section 2, we introduce the network model. In Section 3, we discuss the specific scheme of the D2D social selection relay algorithm combined with the auction principle (SRSA). In Section 4, the proposed algorithm is compared with the existing algorithms through a simulation. In Section 5, the proposed algorithm is summarized.

## 2. D2D Communication System Model

We consider a single-cell scenario, as shown in Figure 1. It is assumed that the cell has one base station, L ideal relays, N pairs of D2D users, and X cellular users. The D2D user includes SUE (Source User Equipment) and DUE (Destination User Equipment). Suppose the sets of SUE and DUE are represented as S={1,2,…,N} and D={1,2,…,N}, respectively, and the SUE and DUE pass through the RUE (Relay User Equipment) for information exchange, the set of RUE is represented by R={1,2,…,L}, and the set of CUE (Cellular User Equipment) is represented by C={1,2,…,X}.

When the quality of the link between the D2D transmitter and the D2D receiver is poor, communication can only be performed through the relay, and each cellular user is pre-allocated with orthogonal channels. During the relay process, the D2D link multiplexes the cellular uplink spectrum resources.

During the D2D communication, we only consider cellular user interference and system noise interference and assume that the location and transmit power of cellular users are fixed. At this time, in the first and second hop links of the D2D relay communication, the SINR (signal-to-interference-plus-noise Ratio) of the double-hop link is $\gamma_{sr}$ and $\gamma_{rd}$ which must be greater than the link threshold $\gamma_{th}$.
(1)$$\gamma_{sr}=\frac{P_{s}{\left|H_{s,r}\right|}^{2}}{P_{c}{\left|H_{c,r}\right|}^{2}+N_{0}}\geq \gamma_{th},$$
(2)$$\gamma_{rd}=\frac{P_{r}{\left|H_{r,d}\right|}^{2}}{P_{c}{\left|H_{c,d}\right|}^{2}+N_{0}}\geq \gamma_{th}$$

At the same time, the communication quality of cellular users must also be guaranteed. That is, (3) and (4) are satisfied.
(3)$$\gamma_{cb}^{s}=\frac{P_{c}{\left|H_{c,b}\right|}^{2}}{P_{s}{\left|H_{s,b}\right|}^{2}+N_{0}}\geq \gamma_{th_{2}},$$
(4)$$\gamma_{cb}^{r}=\frac{P_{c}{\left|H_{c,b}\right|}^{2}}{P_{r}{\left|H_{r,b}\right|}^{2}+N_{0}}\geq \gamma_{th_{2}},$$
where $\gamma_{th}$ and $\gamma_{th2}$ are the SINR thresholds of D2D communication link and cellular communication link, $\gamma_{sr}$ and $\gamma_{rd}$ are the SINR values of the first and second hops of the D2D communication link, $\gamma_{cb}^{s}$ and $\gamma_{cb}^{r}$ are the SINR values of the two phases of the cellular communication link, $P_{s}$ and $P_{r}$ are the transmitted power of node SUE and RUE, $P_{c}$ is the transmit power of node CUE, $H_{s,r}$ and $H_{c,r}$ are the channel gain of SUE to RUE and CUE to RUE, respectively, $H_{r,d}$ and $H_{c,d}$ are the channel gain of RUE to DUE and CUE to DUE, respectively, $H_{c,b}$ and $H_{s,b}$ are the channel gain of CUE to base station, respectively, and $H_{r,b}$ is the channel gain of RUE to base station.

## 3. SRSA Algorithm

### 3.1. Relay Selection Based on Social Threshold

The problem of improving the communication quality of edge users can be transformed into the throughput problem of the link where the user communicates. According to Shannon’s formula, the link throughput is
(5)$$R=Wlog_{2}\left(1+\min\left(\gamma_{sr},\gamma_{rd}\right)\right),$$
subject to
(6)$$\gamma_{cb}^{s},\gamma_{cb}^{r}\geq \gamma_{th_{2}},$$
(7)$$\gamma_{sr}\geq \gamma_{th},$$
(8)$$\gamma_{rd}\geq \gamma_{th}$$

If the above minimum signal-to-interference-to-noise ratio constraints are not met, the D2D communication will be interrupted. D2D first hop outage probability is:(9)$$P_{out}^{1st}=P_{r}\left(\frac{P_{s}{\left|H_{s,r}\right|}^{2}}{P_{c}{\left|H_{c,r}\right|}^{2}+N_{0}}<\gamma_{th}\right),$$

The D2D second hop outage probability is
(10)$$P_{out}^{2nd}=P_{r}\left(\frac{P_{r}{\left|H_{r,d}\right|}^{2}}{P_{c}{\left|H_{c,d}\right|}^{2}+N_{0}}<\gamma_{th}\right),$$

The outage of the entire D2D communication link happens when:The first hop of the D2D link is interrupted.Or the first hop of the D2D link succeeds, but the second hop of the D2D link is interrupted.

So, the total outage probability:(11)$$P_{out}={\;P}_{r}\left(\frac{P_{s}{\left|H_{s,r}\right|}^{2}}{P_{c}{\left|H_{c,r}\right|}^{2}+N_{0}}<\gamma_{th}\right)+{\;P}_{r}\left(\frac{P_{s}{\left|H_{s,r}\right|}^{2}}{P_{c}{\left|H_{c,r}\right|}^{2}+N_{0}}\geq \gamma_{th}\right)\;.P_{r}\left(\frac{P_{r}{\left|H_{r,d}\right|}^{2}}{P_{c}{\left|H_{c,d}\right|}^{2}+N_{0}}<\gamma_{th}\right).$$

When D2D communication is interrupted, the probability of outage from SUE to RUE and RUE to DUE is equal to the cumulative distribution function of SINR of the received signal, i.e.,
(12)$$P_{out}^{1st}=P_{r}\left(\frac{P_{s}{\left|H_{s,r}\right|}^{2}}{P_{c}{\left|H_{c,r}\right|}^{2}+N_{0}}<\gamma_{th}\right)=\int_{0}^{\gamma_{th}}f_{rc}\left(\gamma_{sr}\right)d_{\gamma_{sr}},$$
(13)$$P_{out}^{2nd}=P_{r}\left(\frac{P_{r}{\left|H_{r,d}\right|}^{2}}{P_{c}{\left|H_{c,d}\right|}^{2}+N_{0}}<\gamma_{th}\right)=\int_{0}^{\gamma_{th}}f_{rc}\left(\gamma_{rd}\right)d_{\gamma_{rd}},$$
where $f_{rc}\left(\gamma_{sr}\right)\;\mathrm{and}\;f_{rc}\left(\gamma_{rd}\right)$ are the probability density. The cumulative distribution function of $\gamma_{sr}$ and $\gamma_{rd}$ can be obtained by using lemma 1 in [25]; therefore, the outage probabilities from SUE to RUE and RUE to DUE can be obtained as follows:(14)$$P_{out}^{1st}=P_{r}\left(\frac{P_{s}{\left|H_{s,r}\right|}^{2}}{P_{c}{\left|H_{c,r}\right|}^{2}+N_{0}}<\gamma_{th}\right)=1-\frac{P_{s}}{P_{c}\gamma_{th}+P_{s}}\exp\left(-\frac{\gamma_{th}N_{0}}{P_{s}}\right),$$
(15)$$\;P_{out}^{2nd}=P_{r}\left(\frac{P_{r}{\left|H_{r,d}\right|}^{2}}{P_{c}{\left|H_{c,d}\right|}^{2}+N_{0}}<\gamma_{th}\right)=1-\frac{P_{r}}{P_{c}\gamma_{th}+P_{r}}\exp\left(-\frac{\gamma_{th}N_{0}}{P_{r}}\right).$$

The reference [18] mentions the restricted conditions of D2D direct link, which can be obtained as $\gamma_{th}N_{0}{}^{2}\ll P_{s}$ and $\gamma_{th}N_{0}{}^{2}\ll P_{r}$, $\gamma_{th}$ is usually small, thus
(16)$$\frac{\gamma_{th}N_{0}}{P_{s}}\to 0,\frac{\gamma_{th}N_{0}}{P_{r}}\to 0,$$

Then, (16) is used to simplify the outage probability of D2D communication.
(17)$$P_{out}=1-\frac{P_{s}}{P_{s}+\gamma_{th}P_{c}}\cdot \frac{P_{r}}{P_{r}+\gamma_{th}P_{c}}.$$

From (17), the outage probability $P_{out}$ is inversely proportional to the relay transmission power $P_{r}$, where $P_{r}$ is the ideal power. In this paper, social intention is introduced. The larger $S_{i,j}\;$($0\leq S_{i,j}\leq 1$) is, the stronger social intention is. Therefore, the actual transmission power is $S_{i,j}P_{r}$. Then,
(18)$$P_{out}=1-\frac{P_{s}}{P_{s}+\gamma_{th}P_{c}}\cdot \frac{S_{i,j}P_{r}}{S_{i,j}P_{r}+\gamma_{th}P_{c}}.$$

To reduce the detection cost, a social threshold is introduced to filter the relay set. Social weights and social thresholds are shown in (19) and (20).
(19)$$S_{i,j}=\frac{T_{i,j}}{\sum_{k=1}^{L}T_{i,k}}\cdot \frac{n_{i,j}}{\sum_{k=1}^{L}n_{i,k}},$$
(20)$$w_{i}^{\prime }=\frac{\sum_{k=1}^{L}S_{i,k}}{L},$$
where $T_{i,j}$ is the talk time between SUE i and RUE *j*, $T_{i,k}$ is the total talk time between SUE i and all RUEs, $n_{i,j}$ is the number of calls between SUE i and RUE *j*, $n_{i,k}$ is the number of calls between SUE i and all RUEs. $S_{i,j}$ is the social weight, $w_{i}^{\prime }$ is the social threshold, when $S_{i,j}\geq w_{i}^{\prime }$, the node satisfies the communication social conditions, and the node is added to the candidate relay set Q. Otherwise, delete the node.

After excluding nodes with smaller social weight $S_{i,j}$ according to the social threshold $w_{i}^{\prime }$, the candidate relay set Q is obtained. If there are many relay nodes in the set Q, the relay nodes with smaller outage probability can enter the set H.

### 3.2. Relay Selection Combined with Auction

After screening out the candidate relay set H based on the basis of the previous section, QoS analysis is performed on each candidate relay node. It can be known from (2) and (4). When the relay device performs cooperative communication to satisfy the minimum SINR requirement of communication, the transmit power range of the relay device is:(21)$$P_{r}\geq \frac{\gamma_{th}\left(P_{c}{\left|H_{c,d}\right|}^{2}+N_{0}\right)}{{\left|H_{r,d}\right|}^{2}},$$
(22)$$P_{r}\leq \frac{P_{c}{\left|H_{c,b}\right|}^{2}-\gamma_{th_{2}}N_{0}}{\gamma_{th_{2}}{\left|H_{r,b}\right|}^{2}}.$$

Because of the introduction of social weight $S_{i,j}$, the actual transmission power is $S_{i,j}P_{r}$, and Equations (21) and (22) can be rewritten as:(23)$$S_{i,j}P_{r}\geq \frac{\gamma_{th}\left(P_{c}{\left|H_{c,d}\right|}^{2}+N_{0}\right)}{{\left|H_{r,d}\right|}^{2}},$$
(24)$$S_{i,j}P_{r}\leq \frac{P_{c}{\left|H_{c,b}\right|}^{2}-\gamma_{th_{2}}N_{0}}{\gamma_{th_{2}}{\left|H_{r,b}\right|}^{2}}.$$

From Formula (18), $P_{out}$ decreases with the increase in $S_{i,j}P_{r}$. The social weight $S_{i,j}$ of the relay nodes in H has been fixed. If you want to reduce the outage probability $P_{out}$ again, you can only control the size of the relay transmission power $P_{r}$. Therefore, this paper introduces an auction strategy based on monetary incentives. Monetary incentives encourage relay users to increase their transmit power.

During the auction, D2D users act as holders of monetary resources, and candidate relays make bids based on the offered $P_{r}$ size. D2D users select the idle user with the lowest bid as the relay user. The auction algorithm process is as follows:

(1) Bidding: RUE bids according to the power $P_{r}$ it can provide. Bids between RUEs are confidential and can only be bid once. A seller’s bid is expressed in the following form:(25)$$C_{i}=\lambda_{i}\;P_{r_{i}},$$
where $\lambda_{i}$ is the floor price proposed by relay i in H, and $P_{r_{i}}$ is the cooperative power provided by the relay node i.

(2) Transaction: D2D users want to have as much money left over as possible after paying the relay node currency, that is, the principle of the lowest bid. As shown in the following formula:(26)$$C_{max}=C_{s}-C_{i},$$
where $C_{s}$ is the amount of currency held by D2D users, $C_{i}$ is the price proposed by relay i in (25), the larger $C_{max}$, the higher the satisfaction of D2D users, and the node is the best relay.

The entire relay algorithm process is as follows (Algorithm 1):
**Algorithm 1 Calculate**Initialize all parameters: Pc,Pr,γth,  γth2,  N0For *i* = 1:L Compute: Si,j for each RUE, wi′ If Si,j≥wi′ Then Enter set Q   According to Formula (18), get Pout, Select the node with small Pout to enter the set H End if For *i* = 1:H BringCi into Cmax=Cs−Ci  If Cmax is the largest, i is the optimal relay node End if End forEnd for

## 4. Simulations

In this section, we evaluate and compare the performance of the proposed SRSA algorithm and the existing D2D relay algorithms, including the Random Relay Selection (RRS) and Maximum Throughput-based Relay Selection (MTRS) [26]. A single cellular cell environment is considered, and the geographical location of the users in the cell is subject to random distribution. The channel model considers the path loss [27]. The power mentioned in this paper includes the path loss. The simulation parameters are shown in Table 1.

Figure 2 compares the average detection times of each algorithm with the same number of relays. As can be seen from the figure, the average detection times of the SRSA algorithm are lower than the MTRS algorithm but higher than the RRS algorithm. The reason is that the SRSA algorithm proposes a social threshold $w_{i}^{\prime }$. If the social weight $S_{i,j}\geq w_{i}^{\prime }$, the relay nodes with a weak cooperation intention can be excluded, and the number of probes can be reduced.

From Figure 3, with the increase in idle users, the SRSA algorithm greatly improves the total throughput of the system compared with the RRS algorithms. Our algorithm introduces an auction algorithm to improve the actual transmit power according to the inverse proportion between the outage probability $P_{out}$ and the actual power $S_{i,j}P_{r}$. It not only reduces the outage probability but also improves the SINR of the D2D link. Thus, the SRSA algorithm improves the communication link throughput.

As can be seen from Figure 4, the outage probability of the SRSA algorithm proposed in this paper is better than the MTRS algorithm and RRS algorithm, because the nodes that are not willing to cooperate are excluded by the condition $S_{i,j}\geq w_{i}^{\prime }$. Then, according to the inverse relation between $P_{out}$ and $S_{i,j}P_{r}$, an auction algorithm is introduced to improve the size of $S_{i,j}P_{r}$. Therefore, the outage probability is minimal. However, both the RRS and MTRS algorithms do not consider the user’s willingness to cooperate.

Figure 5 compares the effect of the transmit power on the outage probability. When the number of standby relays is the same, with the increase in the relay node transmit power, the probability of a system outage decreases; when the transmit power is the same, with the increase in the number of standby relays, the probability of a system outage is also reduced. It is consistent with the formula derived in this paper. Therefore, the SRSA algorithm can reduce the outage probability to some extent.

## 5. Conclusions

In this paper, we propose a D2D relay selection algorithm based on an auction strategy. From the social level, the cooperative willingness of relay device users is considered. The stronger the social relationship, the stronger the cooperative willingness and the higher the relay transmission power. Then, we deduced the relationship between the transmission power and the outage probability. The higher the transmission power, the smaller the outage probability. Then, an auction strategy was introduced to encourage relay devices to provide a greater transmission power through monetary incentives. The results show that this algorithm is superior to other relay algorithms; it not only improves the throughput of the system but also reduces the probability of a communication outage.

## Figures and Tables

**Figure 1 sensors-22-09265-f001:**
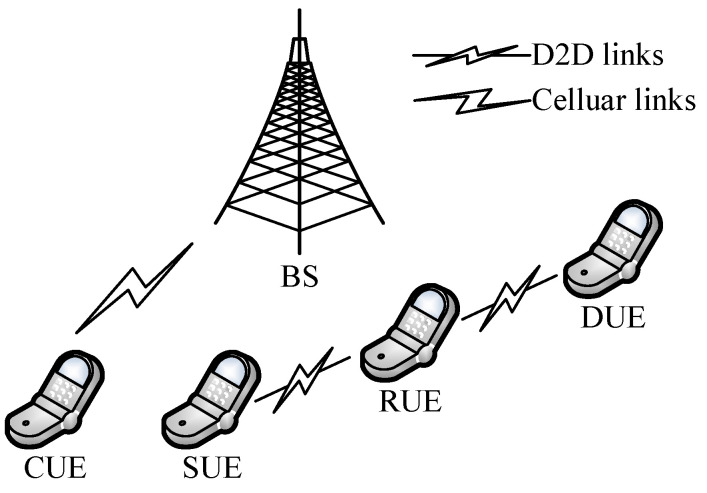
System model.

**Figure 2 sensors-22-09265-f002:**
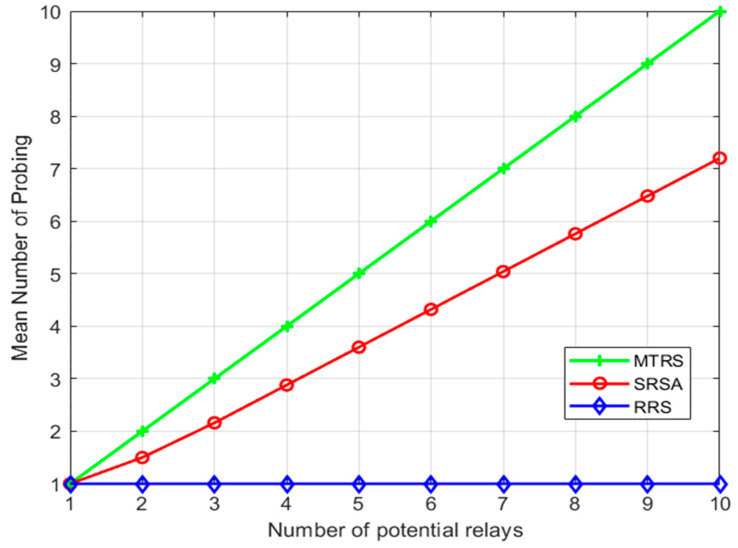
Average detection times of each algorithm.

**Figure 3 sensors-22-09265-f003:**
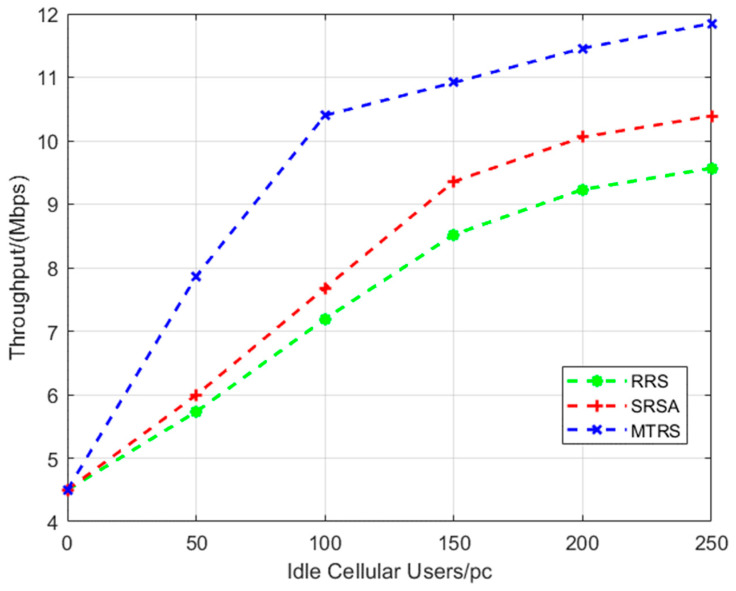
Variation curve of outage probability with the number of idle cellular users.

**Figure 4 sensors-22-09265-f004:**
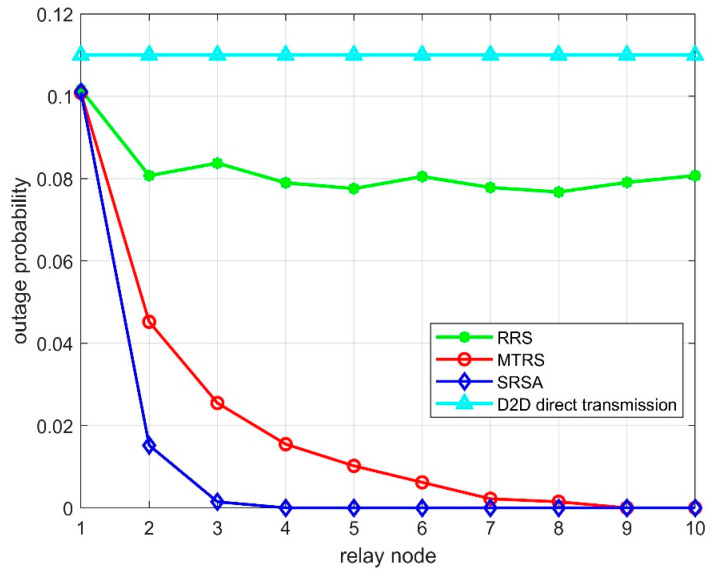
The relationship between the system outage probability and the number of relays.

**Figure 5 sensors-22-09265-f005:**
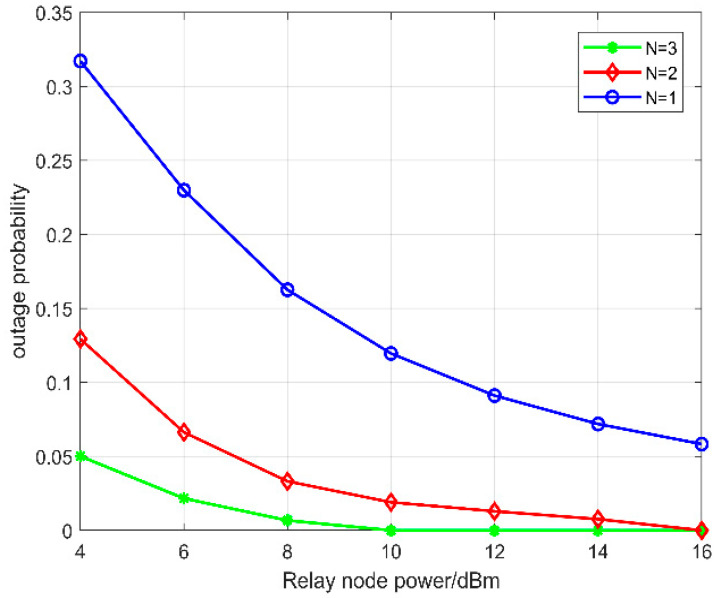
The relationship between system outage probability and relay power.

**Table 1 sensors-22-09265-t001:** Simulation Parameters.

Parameter	Settings
Cellular layout	one isolated cellular cell
Cell radius	500 m
Bandwidth	180 kHz
Noise density	−174 dBm/Hz
Cellular link path loss	128 + 37.6$log_{10}\left(d\left[km\right]\right)$
D2D link path loss	148 + 40$log_{10}\left(d\left[km\right]\right)$
D2D SINR threshold	3 dB
Cellular SINR threshold	5 dB
Number of relay devices	10–100
D2D user-transmitted power	20 dBm
Cellular transmission power	30 dBm

## Data Availability

Not applicable.

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
