# Peer review of "D2D Social Selection Relay Algorithm Combined with Auction Principle"

_sensors, 2022, doi:10.3390/s22239265_

Round 1

Reviewer 1 Report (Previous Reviewer 2)

The authors has revised the paper and responded appropriately to the review questions. I think the paper can be accepted for publishing.

Author Response

Dear Reviewer:

Reviewer 2 Report (Previous Reviewer 3)

The authors have replied to all the queries and concerns with proper references, related to the first submission. However, to improve the connectivity for readers, authors can develop their research in future articles by using a more practical approach. Anyway, the article can use some more improvement, but according to the author's expertise, the paper is properly presented now.

Good luck with future research!

Author Response

Dear Reviewer:

This manuscript is a resubmission of an earlier submission. The following is a list of the peer review reports and author responses from that submission.

Round 1

Reviewer 1 Report

1. references are missing in the paper.

2. The purpose of this paper is unclear.

3. in the paper the experimental part is missing

4. the content needs a lot of development and comparisons

Reviewer 2 Report

A D2D relay selection algorithm is studied in this paper.

There are some problems to be discussed:

1. In the case of multiple relays, the communication throughput will decrease with the increase of relay series, which is not considered in this paper.

2. The proposed scheme only limits the minimum transmit power, but when the transmit power of relay nodes increases, some intermediate relay nodes may not be needed.

3. The paper does not consider the energy efficiency.

4. There are some errors in the paper, for example: the content of line 31-33 is repeated; in Table 1, some parameters are wrong, etc..

Reviewer 3 Report

From the article title point of view, the paper is well-intentioned in D2D (device-to-device) communication. Congrats to the authors for aiming this research. 

State of the art is related to the current concern of the nowadays solution and the research interest area. Of references, there is more than 80 percent from recent research.

There is good mathematical support for the algorithm. What is the simulation software, and how the results can be verified? Does was adopt any error acceptance? 

Is not clear how can a probability be diminished without changing the events, please can be more explicit.

As a recommendation, for future research, try to deploy an experimental test by using professional devices, then compare it to your results from the present research, and then you will understand the difference.  

At line 259, what figure do you refer?

At line 263 the author states as a singular ”… is not as good as the algorithm I proposed.”  this is not ok, you must prove and explain why is not that good the other… . Please reformulate this sentence. If you exprim yourself as singular then why are 4 authors?

Maybe if the article is reconsidered, it can be useful for future research. It does have an interesting idea for considering the auction strategy-based D2D relay selection algorithm.

Maybe the paper can be published in the journal with some major adjustments.